# Genetic Variation in the ASTN2 Locus in Cardiovascular, Metabolic and Psychiatric Traits: Evidence for Pleiotropy Rather Than Shared Biology

**DOI:** 10.3390/genes12081194

**Published:** 2021-07-31

**Authors:** Olivia Burt, Keira J. A. Johnston, Nicholas Graham, Breda Cullen, Donald M. Lyall, Laura M. Lyall, Jill P. Pell, Joey Ward, Daniel J. Smith, Rona J. Strawbridge

**Affiliations:** 1Institute of Health and Wellbeing, University of Glasgow, Glasgow G12 8QQ, UK; 2235998B@student.gla.ac.uk (O.B.); 2340814J@student.gla.ac.uk (K.J.A.J.); Nicholas.Graham@glasgow.ac.uk (N.G.); breda.cullen@glasgow.ac.uk (B.C.); donald.lyall@glasgow.ac.uk (D.M.L.); Laura.Lyall@glasgow.ac.uk (L.M.L.); jill.pell@glasgow.ac.uk (J.P.P.); Joey.Ward@glasgow.ac.uk (J.W.); d.smith@ed.ac.uk (D.J.S.); 2Division of Psychiatry, University of Edinburgh, Edinburgh EH10 5HF, UK; 3School of Life Sciences, College of Medical, Veterinary & Life Sciences, University of Glasgow, Glasgow G12 8QQ, UK; 4Health Data Research UK, Glasgow G12 8QQ, UK; 5Cardiovascular Medicine Unit, Department of Medicine Solna, Karolinska Institutet, 17177 Stockholm, Sweden

**Keywords:** *ASTN2*, cardiovascular disease, metabolic disease, psychiatric illness, anhedonia, mood instability, neuroticism, BMI, central obesity, blood pressure

## Abstract

Background: The link between cardiometabolic and psychiatric illness has long been attributed to human behaviour, however recent research highlights shared biological mechanisms. The *ASTN2* locus has been previously implicated in psychiatric and cardiometabolic traits, therefore this study aimed to systematically investigate the genetic architecture of *ASTN2* in relation to a wide range of relevant traits. Methods: Baseline questionnaire, assessment and genetic data of 402111 unrelated white British ancestry individuals from the UK Biobank was analysed. Genetic association analyses were conducted using PLINK 1.07, assuming an additive genetic model and adjusting for age, sex, genotyping chip, and population structure. Conditional analyses and linkage disequilibrium assessment were used to determine whether cardiometabolic and psychiatric signals were independent. Results: Associations between genetic variants in the ASTN2 locus and blood pressure, total and central obesity, neuroticism, anhedonia and mood instability were identified. All analyses support the independence of the cardiometabolic traits from the psychiatric traits. In silico analyses provide support for the central obesity signal acting through *ASTN2*, however most of the other signals are likely acting through other genes in the locus. Conclusions: Our systematic analysis demonstrates that *ASTN2* has pleiotropic effects on cardiometabolic and psychiatric traits, rather than contributing to shared pathology.

## 1. Introduction

Individuals with serious mental illnesses (SMI, such as schizophrenia, bipolar disorder, and major depressive disorder) suffer from physical diseases at a higher frequency than unaffected individuals and have excess mortality to reflect this—the life expectancy is 25 years shorter on average than the general population, largely attributable to cardiovascular disease [1,2]. Non-genetic factors are commonly used to explain the co-existence between mental and cardiometabolic health conditions, such as lifestyle factors, illness-related factors, and treatment-related factors. For example, individuals with depressive disorders are more likely to smoke and to be obese than non-depressed individuals [1,3]. However, with an increasing understanding of genetics, new theories are emerging that link SMI and physical health on a molecular level, and introduce the possibility of shared genetic predisposition [4].

Through numerous genome-wide association studies (GWAS), the *ASTN2* locus has been identified as associated with both cardiometabolic and psychiatric traits. Psychiatric associations with *ASTN2* single nucleotide polymorphisms (SNPs) include those with bipolar disorder (BPD) [5,6], major depressive disorder (MDD) [7,8,9], and response to anti-psychotic treatment [10]. Cardiometabolic associations include blood pressure [11,12,13], measures of obesity [13,14,15,16] and glucose homeostasis [17]. The *ASTN2* gene encodes astrotactin, a membrane protein [18] implicated in synaptic function [19]; thus, while a role in psychiatric traits is logical, how the *ASTN2* locus might influence cardiometabolic traits is less clear.

We set out to systematically investigate in primary data whether genetic variation in the *ASTN2* locus could contribute to the shared genetic predisposition and potentially shared pathology of cardiometabolic and psychiatric illness, using the UK Biobank dataset. 

## 2. Materials and Methods

### 2.1. Cohort and Phenotyping

The UK Biobank study has been described in open-access publications [20,21,22]. Briefly, ~500,000 volunteers were recruited to the UK Biobank study between 2006–2010, from 22 centres around the UK. At baseline, all individuals underwent a physical examination and completed extensive questionnaires on personal and family medical history and lifestyle. Additional mental health questionnaires and imaging have been completed by a subset of individuals in follow-up requests for data. At baseline, a blood sample was collected and stored appropriately for DNA extraction (buffy coat fraction in EDTA tubes, stored at −80°C (https://www.ukbiobank.ac.uk/media/gnkeyh2q/study-rationale.pdf, accessed on 28 July 2021) and for biomarker analysis. DNA extraction enabled genome-wide genotyping of all individuals which was conducted using 2 genotyping platforms, with a large proportion of overlapping SNPs [20]. Quality control and imputation of genetic data was conducted centrally by UK Biobank, using standard protocols [20]. This study was covered by the generic ethical approval for UK Biobank studies from the NHS National Research Ethics Service (approval letter dated 29 June 2021, Ref 21/NW/0157). This project was completed under UK Biobank applications 6553 (PI. RJS) and 7155 (PI. JPP).

Anthropometric measurements of height, weight, waist and hip circumferences were used to calculate body mass index (BMI, reflective of total obesity) and waist:hip ratio adjusted for BMI (WHRadjBMI, reflecting central adiposity, as per Shungin et al. [23]). Systolic and diastolic blood pressure (SBP and DBP respectively) were measured twice, and the average values were adjusted (as per Ehret et al. [24]: SBP + 15 mmHg and DBP + 10 mmHg if an individual reported use of an anti-hypertensive medication) to correct for anti-hypertensive medication effects. Type 2 diabetes was defined according to Eastwood et al. [25] based on self-report or anti-diabetic medication. Ischemic heart disease (ISH) and Stroke were defined by self-report (#6150, Heart attack/Angina and Stroke, respectively). Venous thromboembolism was defined by self-report (for deep-vein thrombosis and/or pulmonary embolism, #6152). The baseline questionnaire was used to assess risk-taking behaviour (#2040, “Would you describe yourself as someone who takes risks?”) and mood instability (#1920, “Does your mood often go up and down?”), neuroticism (#20127), and anhedonia (#2060, “over the past two weeks, how often have you had little interest or pleasure in doing things?”). For anhedonia, individuals responding “not at all” were considered controls whilst individuals responding with “several days”, “more than half the days” or “nearly every day” were considered cases. A follow-up “thoughts and feelings” questionnaire (2016–2017) was used to assess lifetime MDD, BD and generalised anxiety disorder (GAD) [26]. Individuals responding “don’t know” or “prefer not to say” were excluded. Current smoking was defined by baseline self-report (#20116) with never and former smokers being combined. In a subset of individuals, carotid intima media thickness (IMT) was measured by ultrasound as previously described [27]. Briefly, a single point in the carotid artery was measured at two projections and the mean and maximum values were calculated.

### 2.2. Genetic Data

Genetic variation in the *ASTN2* locus (defined as the *ASTN2* gene +/−250 kb, UCSC (https://genome-euro.ucsc.edu/cgi-bin/hgGateway?redirect=manual&source=genome.ucsc.edu, accessed on 14 October 2020) build 37, chromosome 9: 118,935,000–120,428,000 (14 October 2020) was selected. Analyses were restricted to unrelated individuals of white British ancestry and SNPs with minor allele frequencies (MAF) < 1% were excluded.

### 2.3. Statistical and Genetic Analyses

Genetic analyses were conducted using PLINK 1.07 [28], assuming an additive genetic model. WHRadjBMI was analysed without covariates (as these are included in the construction of the phenotype). All other analyses were adjusted for age, sex, population structure, (eight genetic principle components) and genotyping chip. Analyses of ISH and stroke also included lipid-lowering and anti-hypertensive medication as covariates. Plink 1.07 [28], was also used to calculate the number of independent loci (using–indep-pairwise and default parameters), which was used for Bonferroni correction for multiple testing. This gave a threshold of *p* value 5.45 × 10^−5^ for statistical significance. Results were visualised using LocusZoom [29] and Haploview [30]. Descriptive statistics were calculated using Stata 16.1 (StataCorp, College Station, TX, USA).

### 2.4. Conditional Analyses and Assessment of Linkage Disequilibrium

In order to assess (a) whether there were multiple signals for each trait and (b) the independence signals between each trait, we carried out conditional analyses. Here we reran all significant analyses including the lead SNP from each significant trait as a covariate. As a complimentary approach, we used Haploview [30] to visualise the linkage disequilibrium (LD) between the lead SNPs.

### 2.5. In Silico Follow-Up Analyses 

We used the GWAS catalogue (https://www.ebi.ac.uk/gwas/, accessed on 25 April 2021) to search for traits that have previously been associated with the *ASTN2* locus and compared these to our own results. We also investigated the effects of lead SNPs on gene expression levels using GTEx data (https://www.gtexportal.org/home/, accessed on 10 May 2021) and the LDexpress tool (with CEU population and tissues of relevance to psychiatric and cardiometabolic diseases selected (brain, nerve, adipose, artery, skeletal muscle, pancreas, liver and pituitary). Lead SNPs and those with LD > 0.8 R2 and GTEx *p* < 1 × 10^−5^ were considered (https://ldlink.nci.nih.gov/?tab=ldexpress, accessed on 14 October 2020). The most significant SNPs for each gene and tissue were reported. In addition, we investigated the predicted functional effects of all SNPS meeting the significance threshold for any trait using Ensembl Variant Effect Predictor (VEP) (https://asia.ensembl.org/info/docs/tools/vep/index.html, accessed on 10 May 2021).

## 3. Results

The characteristics of the cohort are presented in Table 1.

### 3.1. ASTN2 Genetic Variations in Cardiometabolic Traits

Significant associations (Table 2 and Figure 1A–D) were observed in the ASTN2 locus for SBP (4 SNPs, lead SNP rs4837585-T, β (SE) −0.189 (0.044) *p* = 2.02 × 10^−5^), DBP (35 SNPs, rs4837585-T, −0.119 (0.026) *p* = 3.56 × 10^−6^), BMI (89 SNPs, rs55654527-A, −0.063 (0.011) *p* = 1.92 × 10^−8^) and WHRadjBMI (192 SNPs, rs13283416-G, −0.0008 (0.001) *p* = 4.52 × 10^−9^). In conditional analyses, where the primary SNP from each analysis was included as a covariate, no secondary signals were observed for SBP (Figure 2B), but secondary signals were evident for DBP (rs34432054-T, primary analyses values 0.174 (0.038), *p* = 4.97 × 10^−6^. Figure 3) and BMI (rs415978-A, 0.052 (0.011), *p* = 1.75 × 10^−6^. Figure 4). Whilst two SNPs remain significant when adjusting for the primary and secondary signals for DBP (Figure 3C), these variants were not significant in the original analysis of DBP therefore are not considered to be a tertiary signal. Further conditional analyses including primary and secondary lead SNPs as covariates identified tertiary signals for BMI (rs10491574-T, 0.078 (0.017), *p* = 2.05 × 10^−6^, Figure 4C). No secondary signals were identified for WHRadjBMI (Figure 5B).

### 3.2. ASTN2 Genetic Variations in Psychological and Psychiatric Traits

Significant associations (Table 2 and Figure 1E–G) were also identified for neuroticism (rs579017-T, 0.084 (0.017) *p* = 3.58 × 10^−7^), anhedonia (2 SNPs, rs144850429-T, OR (95% confidence intervals) 1.12 (1.06–1.18) *p* = 3.52 × 10^−5^) and mood instability (38 SNPs, rs13284474-T, 0.98 (0.97–0.99) *p* = 7.42 × 10^−7^). Of note, rs13284474 is both the lead SNP for primary mood instability and tertiary BMI signals. 

Conditional analyses identified secondary (but not tertiary) signals for neuroticism (rs2296672-T, 0.036 (0.008) *p* = 1.28 × 10^−5^, Figure 6B,C) and anhedonia (rs35979833-AG, 1.03 (1.02–1.05), *p* = 4.03 × 10^−5^, Figure 7B,C), whilst secondary and tertiary signals were identified for mood instability (rs79825568-A, 0.98 (0.97–0.99), *p* = 1.46 × 10^−5^ and rs4836751-A, 1.03 (1.02–1.05), *p* = 2.52 × 10^−5^, respectively, Figure 8B,C).
Figure 4Regional plots for associations between *ASTN2* and (**A**) BMI, (**B**) adjusted for the primary SNP for BMI, (**C**) primary and secondary SNPs for BMI, primary SNP for (**D**) SBP/DBP, (**E**) WHRadjBMI, (**F**) neuroticism, (**G**) anhedonia, and (**H**) mood instability. Significance was set at *p*-value < 5.45 × 10^−5^ or −log10 > 4.26 (Horizontal yellow line). LD (r2) with the lead SNP are indicated by colour. Grey indicates that there is no LD information available.
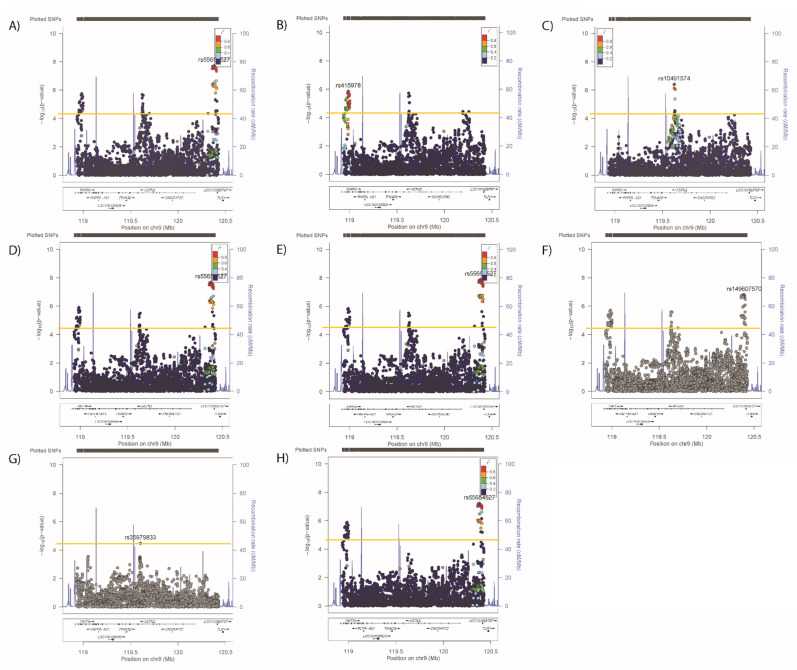

Figure 5Regional plots for associations between *ASTN2* and (**A**) WHRadjBMI, adjusted for the primary SNP for (**B**) WHRadjBMI, (**C**) SBP/DBP, and (**D**) BMI, (**E**) neuroticism, (**F**) anhedonia and (**G**) mood instability. Significance was set at *p*-value < 5.45 × 10^−5^ or −log10 > 4.26 (Horizontal yellow line). LD (r2) with the lead SNP are indicated by colour. Grey indicates that there is no LD information available.
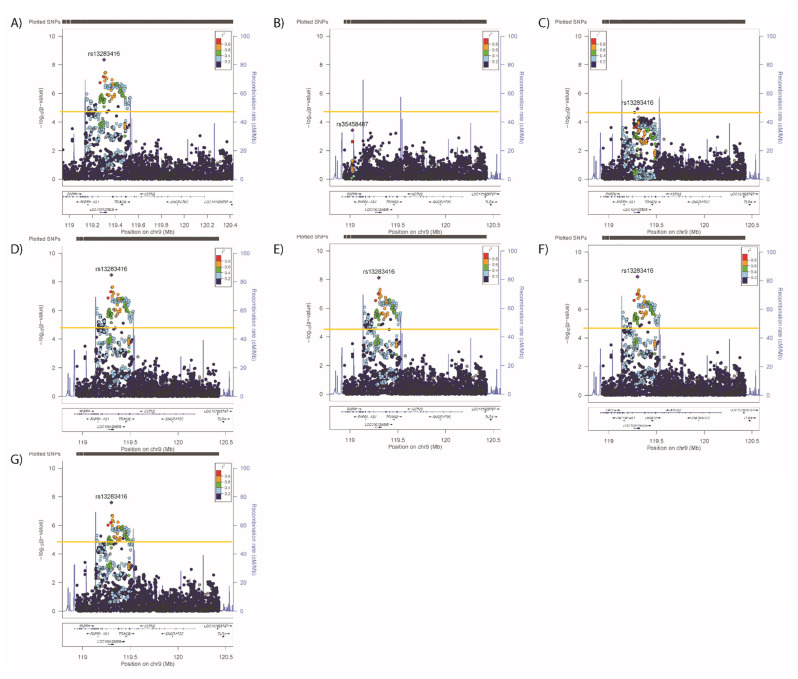



### 3.3. Cross-Trait Observations

The SNPs significantly associated with blood pressure traits or BMI demonstrated consistent effect directions on the other cardiovascular/metabolic traits (Appendix A), so the SBP-increasing allele had a positive effect on DBP and a positive effect on BMI. Because of the way WHRadjBMI is calculated, as an absolute value it is hard to interpret, so it has been omitted from these comparisons. 

SNPs significantly associated with anhedonia, neuroticism or mood instability demonstrated consistent effect directions on the other psychological traits (Appendix A), so the allele associated with increased neuroticism was also associated with increased risk of anhedonia and increased risk of mood instability.

### 3.4. Genetic Architecture of the ASTN2 Locus

Considering the LD between the lead SNPs for all trait signals (Figure 9), it appears that they are largely independent which is consistent with the results of the conditional analyses, with only moderate LD between any two SNPs.

Conditional analyses were also used to assess whether the cardiometabolic signals were independent or overlapped with those for mental health traits. Conditional analyses demonstrated that the lead SNP for SBP remained significant after adjustment for any other primary SNPs (Figure 2C–G), suggesting that the SBP SNP is independent of all other signals. For DBP, adjustment for the other lead signals reduced but did not remove associations of either the primary or secondary signals (Figure 3D–H). For BMI, the primary, secondary and tertiary signals remained significant after adjustment for primary SNPs for blood pressure, WHRadjBMI and neuroticism (Figure 4D–F respectively), however adjusting for the anhedonia primary SNP rendered the primary and secondary signals non-significant (Figure 4G), whilst adjusting for the mood instability primary SNP rendered the tertiary signal non-significant (Figure 4H). Therefore, BMI signals are not completely independent from those for anhedonia and mood instability. When considering WHRadjBMI, adjustment for the primary SBP/DBP SNP reduced but did not remove the association (Figure 5C), whereas there was no effect of adjustment for the other primary signals (Figure 5D–G).

Adjusting neuroticism for the primary signals for SBP/DBP, BMI, WHRadjBMI or anhedonia had no impact on the associations (Figure 6D–G), whilst adjustment for the primary mood instability signal rendered the secondary association non-significant (Figure 6H). Adjusting the anhedonia analyses for the physical traits or neuroticism had no impact on the associations (Figure 7D–H), whilst adjusting for the mood instability primary signal rendered the anhedonia signals null (Figure 7H). The primary and secondary signals for mood instability were uninfluenced by adjusting for the primary SNPs for other traits (Figure 8D–H), however the tertiary mood instability signal was reduced by adjustment for the primary BMI signal (Figure 8D).

### 3.5. Comparison with Previously Published Associations

Robustly reported associations with the *ASTN2* locus are presented in Table 3. Of the lead (primary, secondary or tertiary) SNPs, the lead SNP for WHRadjBMI (rs13283416) has previously been robustly implicated in osteoarthritis [31] and estimated glomerular filtration rate [32,33] whilst the lead SNP for neuroticism (rs579017) has previously been reported for this trait in two studies [34,35]. The current and previous reports of rs579017-T being associated with increased neuroticism are consistent. The allele associated with “poor metabolic health” (increased SBP, DBP, BMI, rs13283416-G) was associated with increased risk of osteoarthritis, which could be expected, however the positive relationship with estimated glomerular filtration rate is unexpected. The association of rs13283416 with multiple components of metabolic syndrome (clustering of increased blood pressures and obesity with dyslipidaemia and diabetes (https://www.nhs.uk/conditions/metabolic-syndrome/, accessed on 28 July 2021) is noteworthy: because of the growing evidence of firstly, a bi-directional relationship between the brain and metabolic systems [36] and secondly, the overlap in genetic regulation of eating disorders such as anorexia nervosa and insulin signalling [37] (widely considered to be the underlying process linking metabolic syndrome components).

SNPs which have previously been associated with either psychological or cardiometabolic traits are presented in Table 3. Of note is the association between rs1861881-G and DBP, which appear discordant between Evangelou et al. [11,12] and our results, however closer inspection of the published results is consistent with our results (suggesting that there is an error with the GWAS catalogue excerpt). Unfortunately, the reasons for the discrepancy between our results for rs17292540 and those reported by Pulit et al. [15] are hard to clarify, as this SNP was not available in that paper’s Appendix A.

With psychiatric traits, ASTN2 has been associated with a range of conditions—bipolar, depression, autism, ADHD and schizophrenia [6,7,8,9]. Our findings are largely consistent with these previous findings (Table 3), in that alleles associated with increased risk of MDD [7,9], BD [5] and ADHD [38] are associated (non-significantly) with increased neuroticism, mood instability and anhedonia (Table 3). ASTN2 has also been associated with migraines and chronic pain- an area known to correlate strongly with mental health [39,40]. Our results are consistent with these, with alleles increasing the risk of chronic pain or migraines being associated with increased neuroticism and mood instability (Table 3).

In addition to GWAS, *ASTN2* has been implicated in SCZ through candidate gene studies, but this is largely through a very rare variant, and has shown variable results in follow-up studies so does not offer much support to the role of ASTN2 in common conditions [40]. Nevertheless, combining the results of all previous studies, there is a strong suggestion of the significance of ASTN2 in psychiatric health, supporting the need for further studies specifically focused on the *ASTN2* locus.

With regards to cardiometabolic health, previous literature again demonstrates several associations between *ASTN2* and the cardiometabolic risk factors mentioned previously. The reported association with increased risk of T2D (rs1885234-G) [41] is consistent with increased SBP (but not DBP) and WHRadjBMI observed here. Notably, a pathway analysis demonstrated a connection between *ASTN2* and plasma triglycerides which further suggested the possible influence on neuronal pathways and insulin sensitivity, contributing to obesity and T2D [42]. The novelty of the mechanism proposed and the relatively small sample size of that study (N = 1000 individuals) means that verification of this association is warranted.

### 3.6. Functional Assessment of ASTN2 Genetic Variants

LD Express and GTEX were used to identify SNPs with genotype-specific effects on gene expression levels in relevant tissues. Results of protein-encoding genes are presented in Table 4 (of note, no data was available for rs415978, rs2296672, rs35979833, rs13284474 and rs79825568). While there were several SNPs with genotype-specific effects on RP11-500B12.1 in brain, nerve, pancreas and pituitary tissues, the significance is unclear, as this hypothetical gene has no known function.

There is limited evidence for ASTN2 being the gene through which the lead SNPs (or their LD proxies) have their effects: a proxy (r2 = 0.92) for the WHRdajBMI lead SNP demonstrated reduced levels of ASTN2 expression with the WHRadjBMI-decreasing allele in subcutaneous adipose tissue (Figure 10A), and a proxy (r2 = 1.0) for the anhedonia lead SNP demonstrated reduced levels of ASTN2 in nerve tissue for the allele that increased risk of anhedonia (Figure 10B). Relaxing the gene expression *p* value to <0.05 meant an additional 3 proxies for the WHRadjBMI lead SNP were significantly associated with ASTN2 levels, suggesting that this result is not due to different sample sizes and power for the different tissues. 

The secondary signal for DBP influences TLR4 in whole blood, with the DBP-increasing allele (rs34432054) associated with reduced levels of TLR4. Likewise, the primary signal for BMI influences TLR4 in whole blood, with the BMI-decreasing allele (rs55654527) being associated with increased levels of TLR4. The primary signal for neuroticism (rs579017) also influences TLR4 in whole blood, with the neuroticism-increasing allele being associated with increased levels of TLR4. Assessment of all SNPs with significant effects on at least one trait for predicted effects on the genes or proteins demonstrated no moderate or severe consequences.

## 4. Discussion

This candidate gene study sought to determine whether the *ASTN2* locus might contribute to shared mechanisms underlying CMD and SMI. We identified associations between genetic variation in *ASTN2* and cardiometabolic traits (specifically SBP, DBP, BMI, WHRadjBMI) as well as traits of relevance to SMI (specifically neuroticism, mood instability and anhedonia). We demonstrated that there is consistency in effects between CMD traits and between SMI traits, but that the relationship between CMD and SMI traits is complex and inconsistent within the locus. 

*ASTN2* is found on chromosome 9 and encodes a transmembrane protein expressed primarily in the brain (Figure 10), largely in the cerebellum [19], therefore it is possible that SNPs in this locus might act through ASTN2 to influence psychological traits. However, it should be noted that the complex cognitive abilities which involve the cerebellum are not necessarily features of the psychiatric traits we have assessed. The protein is known to form a complex with surface membrane protein, ASTN1, and is thought to aid neuronal migration through this interaction [43]. During later stages of development, it is believed that ASTN2 is involved in expression [44] and translocation of a variety of membrane receptors, thereby regulating synaptic functions [19]. Functional analysis has demonstrated reduced electrophysical properties and aberrant connectivity in *ASTN2* knockout neurons [45]. Thus it is plausible that the reduced *ASTN2* levels being associated with the same allele as increases risk of anhedonia suggests a lack synaptic signalling in nerve tissue as a component of anhedonia biology. The fact that multiple signals in the *ASTN2* locus were associated (with consistent effects) with neuroticism, anhedonia, and mood instability might suggest common mechanisms underlying these traits. Indeed, our results agree with numerous studies demonstrating the association between mood instability and neuroticism, which is reassuring, given that mood instability has been described as a “distinctive feature of neuroticism” [46]. Prior studies have implicated this locus in determining psychiatric traits [44,45,47,48] which overlap to varying degrees with these intermediate traits, providing additional weight to the hypothesis that alterations in synaptic signalling could underpin many psychological and psychiatric phenotypes. 

The reduced levels of *ASTN2* being associated with the same allele as reduced central fat accumulation (WHRadjBMI) in adipose tissue is less easy to interpret. The *ASTN2* locus has been associated with triglyceride levels [42] and there is clear crosstalk between the central nervous system and metabolic regulation [49], but the exact role of ASTN2 requires elucidation. Studies of ASTN2 function have focused on neurodevelopmental rather than anthropometric phenotypes [19,44,45] and no murine *astn2* knockout models have been developed. Mice lacking a *brinp1,* a member of the same protein family as *astn2*, which is located nearby *astn2* and has potentially overlapping functions has been investigated [48]. These mice demonstrate reduced postnatal viability and reduced post-natal weight gain [48]. Therefore, *ASTN2* being the gene through which the WHRadjBMI locus acts is reasonable but requires further investigation.

The lack of genotype-specific *ASTN2* expression effects for neuroticism, mood instability, blood pressure and BMI might imply that this is not the functional gene. However, as maturation of the ASTN2 protein requires cleavage of a signal peptide in the endoplasmic reticulum [50], with protein products remaining linked by a disulphide bond [51], genetic variation in the signal peptide, the cleavage site and/or the cysteine residues in the di-sulphide bond would have severe functional effects not visible through mRNA analysis. It also cannot be ruled out that the loci for mood instability, neuroticism, blood pressure and BMI might be acting through other genes in the region, which is supported by their effects on mRNA expression of *TRIM32* and *TLR4* (Table 4). 

It is much harder to draw conclusions on the connection between cardiometabolic and psychiatric health in this locus. Most of our data do not seem to support shared mechanistic contribution to both psychiatric and cardiovascular health. Except for BMI, none of the cardiometabolic lead SNPs are also significant in the psychiatric traits, and conditional analyses as well as the LD structure do not suggest coinheritance of psychiatric and cardiometabolic traits. However, the BMI association could have a behavioural explanation, whereby *ASTN2* influences food preferences as well as mood-related traits, and the food preferences have subsequent effects on BMI: An experimental study in mice has demonstrated that a high fat diet (which causes fat accumulation) can prompt an anhedonia-like phenotype in a manner dependent upon gut microbiota and leptin signalling [52]. There are caveats that should be considered when extrapolating murine studies to real-world human populations. Whilst mice (nocturnal) can be given controlled diets (high fat or normal) and the effect of antibiotics (as a blunt way to change the gut microbiota), behaviour of real-world humans (predominantly diurnal) is more complex, with food preferences, healthcare seeking, medication adherence, physical activity patterns and mood-altering substance use all being interrelated. So whilst this explanation is intriguing, it should be considered with these caveats in mind. Given that the *ASTN2* locus appears to be divided, with one part having consistent direction of effects on CMD and SMI and the other part having opposite effects on CMD and SMI in white British ancestry individuals, further investigation of this locus is warranted

This work was based on the UK Biobank cohort, which is one of the largest studies to date with genetic data as well as a wide range of cardiometabolic, psychological and psychiatric phenotyping. Limitations of the cohort include its bias towards healthier and wealthier individuals [53]. Psychiatric conditions, in particular, are rarer amongst individuals who participated compared to the general population [53]. A cohort with a higher prevalence of cardiometabolic and psychiatric conditions may have allowed identification of further SNPs in the locus which could have been overlooked in this cohort. Similarly, we analysed only individuals with White British ancestry, in order to minimise genetic heterogeneity. Unfortunately, this means results are only generalisable to White British ancestry individuals. 

Future work is required to fully understand the role of the *ASTN2* locus in mental and physical health. Expanding this study to include high psychiatric and/or cardiometabolic risk and/or prevalence populations would be of value, as would extending the study to additional ancestry groups with different LD patterns. In particular, haplotype analyses in a diverse ancestry individuals could elucidate the impact of the consistent/divergent parts of the locus. Additional work in experimental models to better define the role of ASTN2 and effects of reduced levels, particularly in neuronal and adipose tissue would be of interest.

## 5. Conclusions

In summary, this study does not offer any strong evidence for shared genetic mechanisms between cardiometabolic and psychiatric traits in the *ASTN2* locus. However, we do provide evidence for SNPs in this locus acting through *ASTN2* to influence central fat accumulation as well as risk of anhedonia. Whilst we cannot exclude the possibility of a behavioural explanation (as proposed by [52]), the available evidence from experimental studies is more supportive of pleiotropic effects on mental and physical health traits.

## Figures and Tables

**Figure 1 genes-12-01194-f001:**
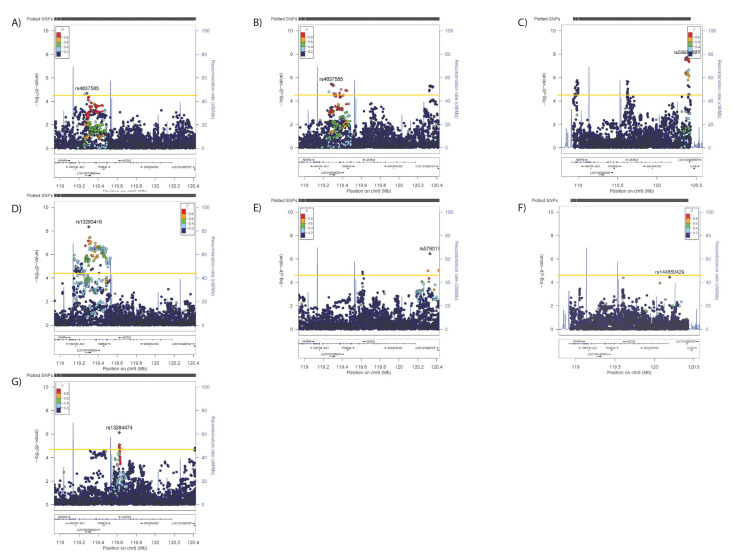
Regional plots for associations between *ASTN2* and (**A**) SBP, (**B**) DBP, (**C**) BMI and (**D**) WHRadjBMI, (**E**) neuroticism, (**F**) anhedonia and (**G**) mood instability. Significance was set at *p*-value < 5.45 × 10^−5^ or −log10 > 4.26 (Horizontal yellow line). LD (r2) with the lead SNP are indicated by colour. Grey indicates that there is no LD information available.

**Figure 2 genes-12-01194-f002:**
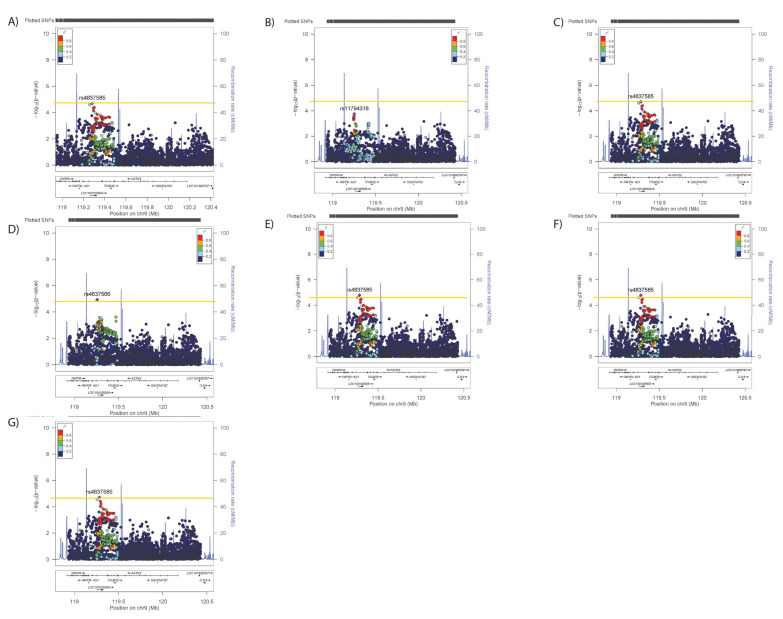
Results of associations between *ASTN2* and (**A**) SBP, adjusted for the primary SNP for (**B**) SBP/DBP, (**C**) BMI and (**D**) WHRadjBMI, (**E**) neuroticism, (**F**) anhedonia and (**G**) mood instability. Significance was set at *p*-value < 5.45 × 10^−5^ or −log10 > 4.26 (Horizontal yellow line). LD (r2) with the lead SNP are indicated by colour. Grey indicates that there is no LD information available.

**Figure 3 genes-12-01194-f003:**
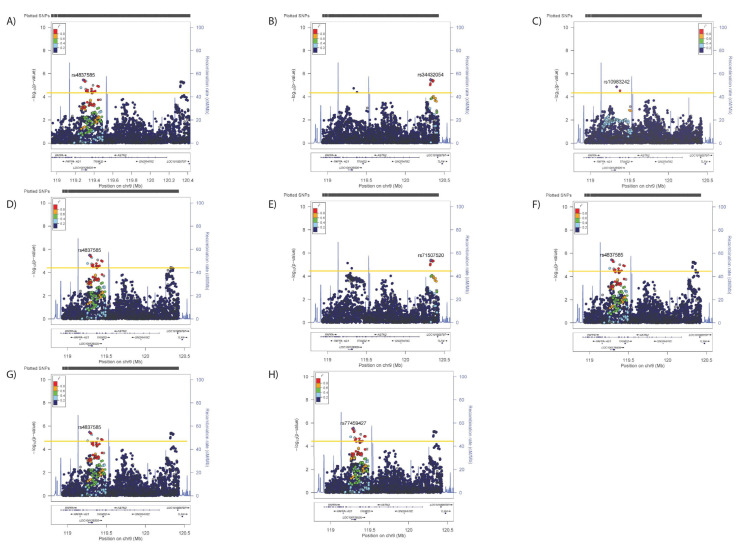
Results of associations between *ASTN2* and (**A**) DBP, adjusted for the primary SNP for (**B**) SBP/DBP, (**C**) primary and secondary SNPs for DBP, (**D**) primary SNP for BMI, (**E**) WHRadjBMI, (**F**) neuroticism, (**G**) anhedonia and (**H**) mood instability. Significance was set at *p*-value < 5.45 × 10^−5^ or −log10 > 4.26 (Horizontal yellow line). LD (r2) with the lead SNP are indicated by colour. Grey indicates that there is no LD information available.

**Figure 6 genes-12-01194-f006:**
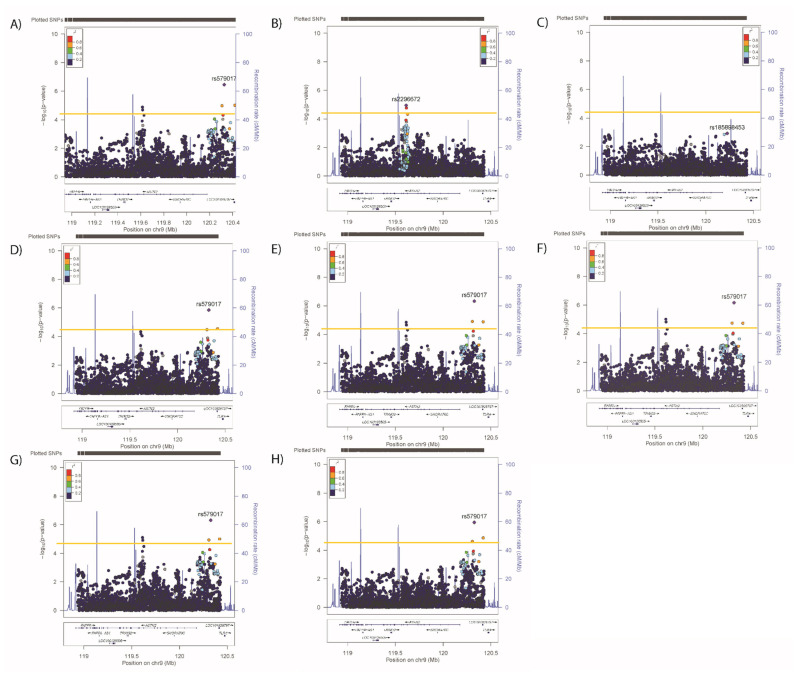
Results of associations between *ASTN2* and (**A**) neuroticism, adjusted for the primary SNP for (**B**) neuroticism, (**C**) primary and secondary SNPs for neuroticism, (**D**) primary SNP for SBP/DBP, (**E**) BMI, (**F**) WHRadjBMI, (**G**) anhedonia and (**H**) mood instability. Significance was set at *p*-value < 5.45 × 10^−5^ or −log10 > 4.26 (Horizontal yellow line). LD (r2) with the lead SNP are indicated by colour. Grey indicates that there is no LD information available.

**Figure 7 genes-12-01194-f007:**
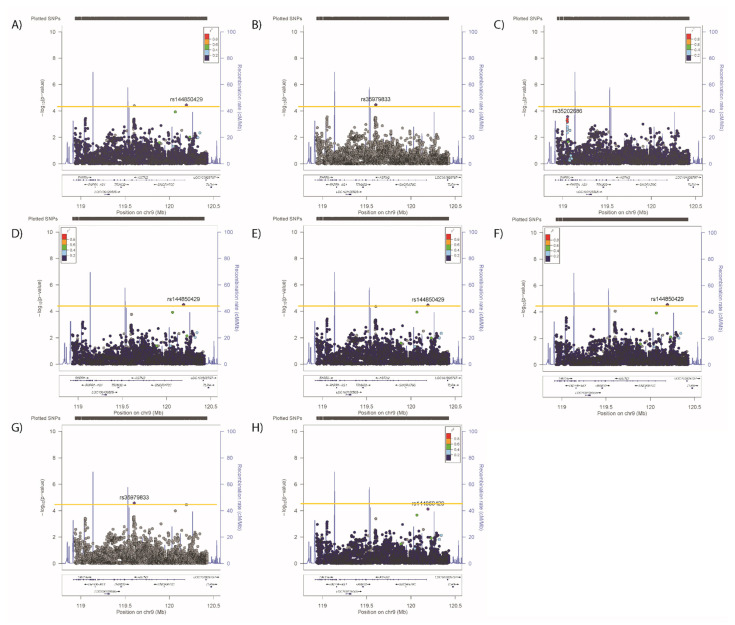
Results of associations between *ASTN2* and (**A**) anhedonia, adjusted for the primary SNP for (**B**) anhedonia, (**C**) primary and secondary SNPs for anhedonia, (**D**) primary SNP for SBP/DBP, (**E**) BMI, (**F**) WHRadjBMI, (**G**) neuroticism and (**H**) mood instability. Significance was set at *p*-value < 5.45 × 10^−5^ or −log10 > 4.26 (Horizontal yellow line). LD (r2) with the lead SNP are indicated by colour. Grey indicates that there is no LD information available.

**Figure 8 genes-12-01194-f008:**
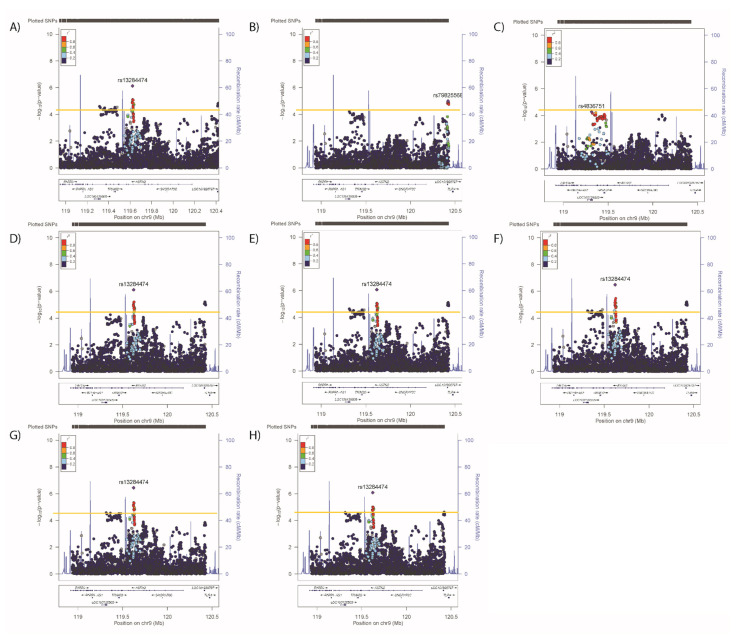
Results of associations between *ASTN2* and (**A**) mood instability, adjusted for the primary SNP for (**B**) mood instability, (**C**) primary and secondary SNPs for mood instability, (**D**) primary SNP for SBP/DBP, (**E**) BMI, (**F**) WHRadjBMI, (**G**) neuroticism and (**H**) anhedonia. Significance was set at *p*-value < 5.45 × 10^−5^ or −log10 > 4.26 (Horizontal yellow line). LD (r2) with the lead SNP are indicated by colour. Grey indicates that there is no LD information available.

**Figure 9 genes-12-01194-f009:**
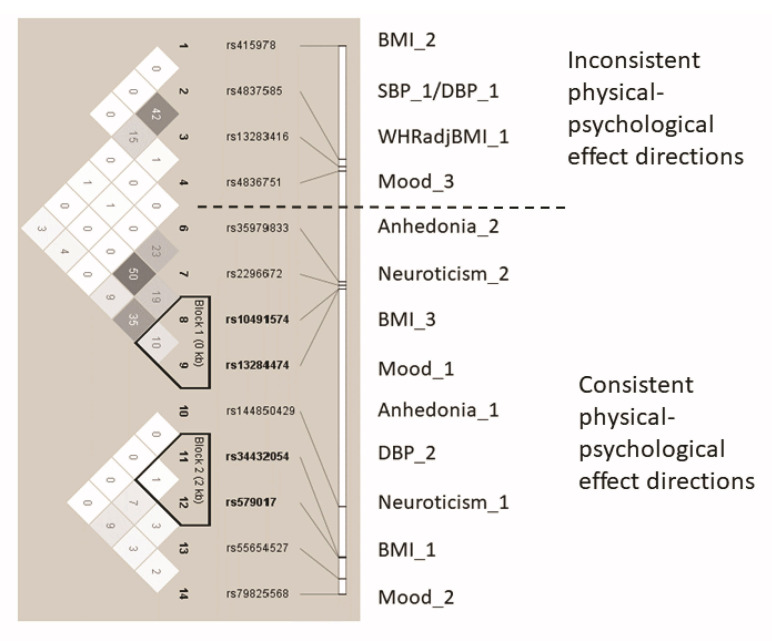
LD Structure of the *ASTN2* locus and associations with psychological and physical phenotypes.

**Figure 10 genes-12-01194-f010:**
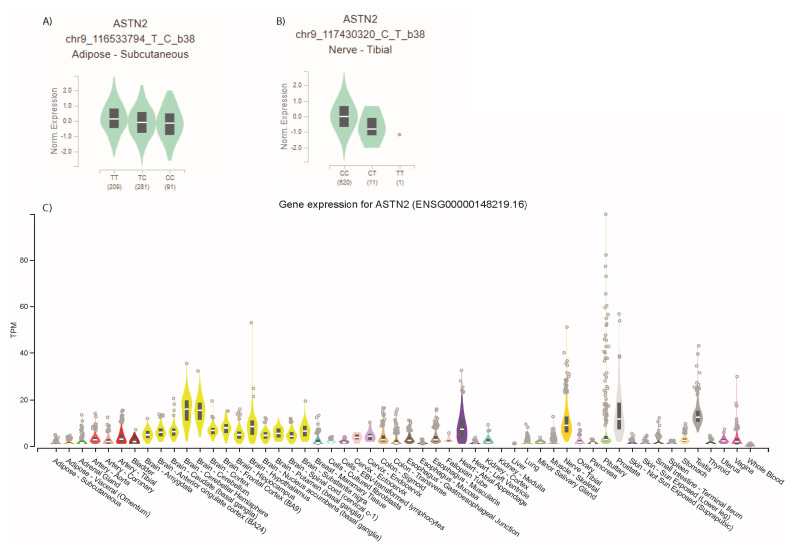
Expression of *ASTN2*, (**A**) depends upon genotype of the WHRadjBMI lead, rs6478243, in adipose tissue, and (**B**) depends on the genotype of the anhedonia lead, rs144850429, in tibial nerve tissue, (**C**) in all tissues.

**Table 1 genes-12-01194-t001:** Descriptions of the UK Biobank participants included in this study.

	Men	Women	All
N (% Male participants)	184,861	217,250	402,111 (46.0)
Age (years)	57.1 (8.1)	56.7 (7.9)	56.9 (8.0)
BMI (kg/m^2^)	27.8 (4.2)	27.0 (5.1)	27.4 (4.8)
WHRadjBMI	0.936 (0.065)	0.817 (0.070)	0.872 (0.09)
SBP (mmHg)	141 (17)	136 (19)	138 (17)
DBP (mmHg)	84 (10)	81 (10)	82 (10)
SBPadj (mmHg)	145 (19)	138 (21)	141 (21)
DBPadj (mmHg)	87 (11)	83 (11)	84 (11)
Hypertension	107,646 (60.9)	97,820 (47.3)	205,466 (53.6)
Anti-hypertensive medication	4328 (2.3)	2770 (1.3)	7098 (1.8)
Lipid-lowering medication	42,600 (25.9)	27,461 (15.6)	70,061 (20.6)
IMTmean *	0.706 (0.135)	0.658 (0.109)	0.681 (0.125)
IMTmax *	0.951 (0.216)	0.874 (0.185)	0.911 (0.204)
IMTmeanmax *	0.823 (0.159)	0.764 (0.131)	0.792 (0.148)
ISH	13,155 (9.8)	5199 (3.1)	18,354 (6.1)
Stroke	3665 (3.0)	2471 (1.5)	6138 (2.1)
Venous thromboembolism	4317 (3.2)	6112 (4.1)	10,429 (3.7)
Type 2 diabetes	11,149 (6.0)	6207 (2.9)	17,356 (4.3)
Current Smoking	21,780 (11.8)	18,769 (8.7)	40,549 (10.1)
Anhedonia	36,860 (19.9)	41,873 (19.3)	78,733 (19.6)
Mood instability	77,034 (41.7)	100,647 (46.3)	177,681 (44.2)
Neuroticism scores	3.58 (3.19)	4.57 (3.24)	4.11 (3.26)
Risk-taking	60,289 (33.6)	37,966 (18.1)	98,255 (25.3)
GAD **	3063 (7.2)	6018 (12.8)	9081 (10.1)
BD **	873 (0.5)	997 (0.5)	1870 (0.5)
MDD **	9661 (5.5)	21,163 (10.3)	30,824 (8.1)

Where continuous variables are presented as mean (standard deviation) and binary variables as N (%). IMT, carotid intima-media thickness; ISH, ischemic heart disease; GAD, generalised anxiety disorder; MDD, major depressive disorder; BD, bipolar disorder. * only available for 22,179 individuals.** only available for 128,751 individuals.

**Table 2 genes-12-01194-t002:** Results of main analyses.

			SBP (N = 367643)		DBP (N = 367646)		BMI (N = 392421)		WHRadjBMI (N = 392319)		Neuroticism (N = 318949)		Anhedonia (N = 379986)		Mood Instability (N = 383367)	
SNP	A1	MAF	β (SE)	*p*	β (SE)	*p*	β (SE)	*p*	β (SE)	*p*	β (SE)	*p*	OR (95%CI)	*p*	OR (95%CI)	*p*
rs4837585 ^a^	T	0.46	−0.189 (0.044)	**2.02 × 10^−5^**	−0.119 (0.026)	**3.56 × 10^−6^**	−0.003 (0.011)	0.7881	0.0005 (0.0001)	0.0001	0.011 (0.008)	0.1741	1.01 (1.00–1.02)	0.0396	1.01 (1.00–1.02)	0.1137
rs34432054 ^b^	T	0.13	0.177 (0.066)	0.0071	0.174 (0.038)	**4.97 × 10^−6^**	0.041 (0.016)	0.0099	−0.0001 (0.0002)	0.4802	−0.004 (0.012)	0.7081	1.01 (0.99–1.03)	0.3627	1.01 (1.00–1.03)	0.0428
rs55654527 ^a^	A	0.34	−0.027 (0.046)	0.5532	−0.060 (0.027)	0.0243	−0.063 (0.011)	**1.92 × 10^−8^**	0.0003 (0.0001)	0.0280	−0.007 (0.008)	0.4235	1.00 (0.99–1.01)	0.8956	1.00 (0.99–1.01)	0.6176
rs415978 ^b^	A	0.39	0.076 (0.045)	0.0916	0.022 (0.026)	0.4006	0.052 (0.011)	**1.75 × 10^−6^**	0.0000 (0.0001)	0.9140	−0.024 (0.008)	0.0029	0.99 (0.98–1.00)	0.0480	1.00 (0.99–1.01)	0.5227
rs10491574 ^c^	T	0.12	0.089 (0.068)	0.1916	0.053 (0.040)	0.1792	0.078 (0.017)	**2.05 × 10^−6^**	0.0003 (0.0002)	0.2048	0.037 (0.012)	0.0027	1.02 (1.01–1.04)	0.0112	1.02 (1.01–1.04)	0.0018
rs13283416 ^a^	G	0.43	0.112 (0.045)	0.0118	0.068 (0.026)	0.0084	0.001 (0.011)	0.9015	−0.0008 (0.0001)	**4.52 × 10^−9^**	−0.003 (0.008)	0.7248	0.99 (0.98–1.00)	0.0504	0.99 (0.99–1.00)	0.2772
rs579017 ^a^	T	0.06	0.003 (0.091)	0.9700	0.023 (0.053)	0.6607	0.049 (0.022)	0.0255	−0.0002 (0.0003)	0.4134	0.084 (0.017)	**3.58 × 10^−7^**	1.01 (0.98–1.03)	0.6639	1.03 (1.01–1.05)	0.0013
rs2296672 ^b^	T	0.37	0.003 (0.046)	0.3958	0.011 (0.026)	0.6695	0.037 (0.011)	0.0009	−0.0001 (0.0001)	0.6593	0.036 (0.008)	**1.28 × 10^−5^**	1.02 (1.00–1.03)	0.0103	1.02 (1.01–1.03)	**4.15 × 10^−5^**
rs144850429 ^a^	T	0.01	0.003 (0.211)	0.5737	0.054 (0.122)	0.6588	0.166 (0.051)	0.0228	−0.0001 (0.0006)	0.8498	0.054 (0.038)	0.1593	1.12 (1.06–1.18)	**3.52 × 10^−5^**	1.03 (0.98–1.07)	0.2359
rs35979833 ^b^	AG	0.15	0.003 (0.063)	0.7300	0.016 (0.036)	0.6519	0.049 (0.015)	0.0014	0.0001 (0.0002)	0.5667	0.035 (0.011)	0.0022	1.03 (1.02–1.05)	**4.03 × 10^−5^**	1.02 (1.01–1.04)	0.0005
rs13284474 ^a^	T	0.45	0.003 (0.044)	0.4141	0.019 (0.026)	0.4581	−0.045 (0.011)	**2.52 × 10^−5^**	−0.0003 (0.0001)	0.0421	−0.022 (0.008)	0.0064	0.99 (0.98–1.00)	0.0189	0.98 (0.97–0.99)	**7.42 × 10^−7^**
rs79825568 ^b^	A	0.48	0.003 (0.044)	0.8172	−0.033 (0.025)	0.1987	−0.016 (0.011)	0.1281	0.0001 (0.0001)	0.6696	−0.023 (0.008)	0.0046	0.99 (0.98–1.00)	0.2343	0.98 (0.97–0.99)	**1.46 × 10^−5^**
rs4836751 ^c^	A	0.12	0.003 (0.069)	0.0710	−0.069 (0.040)	0.0823	0.001 (0.000)	0.0078	0.0202 (0.0166)	0.2221	0.032 (0.012)	0.0110	1.02 (1.00–1.04)	0.0206	1.03 (1.02–1.05)	**2.52 × 10^−5^**

Where: ^a^, primary; ^b^, secondary; ^c^, tertiary; all values are from the primary analyses.

**Table 3 genes-12-01194-t003:** Reported associations between the ASTN2 locus and psychological or cardiometabolic traits.

Published							UK Biobank									
PMID	Trait	Size	Lead SNP	RA	RAF	OR or β	A1	A2	MAF	β SBP	β DBP	β BMI	β WHRadjBMI	β Neuroticism	OR Mood Instability	OR Anhedonia
23793025	Migraine	b	rs17303101	A	0.28	1.07	A	G	0.291	−0.01	0.03	0.00	−0.0003	0.01	1.00	1.00
31676860	Brain region volumes	b	rs10983184				C	T	0.359	−0.07	−0.04	0.01	0.0004	0.00	1.00	1.00
31676860	Brain region volumes	b	rs1040851				C	A	0.424	−0.06	−0.01	0.00	0.0001	0.02	1.01	1.01
31676860	Brain region volumes	b					C	A	0.424	−0.06	−0.01	0.00	0.0001	0.02	1.01	1.01
30279459	Total hippocampal volume	b	rs7873551	C		42.42	C	T	0.231	0.00	0.05	−0.01	0.0000	0.00	1.00	1.00
31676860	Brain region volumes	b	rs7030607				A	G	0.359	0.02	0.01	−0.01	0.0004	−0.02	0.99	1.00
31676860	Brain region volumes	b					A	G	0.359	0.02	0.01	−0.01	0.0004	−0.02	0.99	1.00
27182965	Migraine	b				1.05	A	G	0.359	0.02	0.01	−0.01	0.0004	−0.02	0.99	1.00
32541925	Type 2 diabetes	b	rs1885234	G	0.39	0.02	G	T	0.419	0.01	−0.01	0.00	0.0004	−0.01	1.00	1.01
30279459	Dentate gyrus granule cell layer volume	b	rs6478241	A		3.80	A	G	0.365	−0.04	0.01	−0.01	0.0000	0.02	1.01	1.00
22683712	Migraine	c		A	0.38	1.16	A	G	0.365	−0.04	0.01	−0.01	0.0000	0.02	1.01	1.00
27322543	Migraine	b		A	0.36	1.05	A	G	0.365	−0.04	0.01	−0.01	0.0000	0.02	1.01	1.00
23793025	Migraine–clinic-based	b		A	0.38	1.16	A	G	0.365	−0.04	0.01	−0.01	0.0000	0.02	1.01	1.00
23793025	Migraine without aura	b		A	0.38	1.12	A	G	0.365	−0.04	0.01	−0.01	0.0000	0.02	1.01	1.00
27322543	Migraine without aura	b		A	0.35	1.14	A	G	0.365	−0.04	0.01	−0.01	0.0000	0.02	1.01	1.00
31194737	Multisite chronic pain	b		A	0.37	0.01	A	G	0.365	−0.04	0.01	−0.01	0.0000	0.02	1.01	1.00
31676860	Brain region volumes	b	rs34979631				T	C	0.225	0.01	0.06	−0.01	0.0000	0.01	1.01	1.00
31676860	Brain region volumes	b					T	C	0.225	0.01	0.06	−0.01	0.0000	0.01	1.01	1.00
31676860	Brain region volumes	b	rs4837565				A	G	0.139	0.14	0.10	0.01	0.0003	0.00	1.01	1.00
31676860	Brain region volumes	b					A	G	0.139	0.14	0.10	0.01	0.0003	0.00	1.01	1.00
31676860	Brain region volumes	b	rs11792948				A	G	0.359	0.05	0.05	0.00	0.0001	0.01	1.01	1.00
31676860	Brain region volumes	b					A	G	0.359	0.05	0.05	0.00	0.0001	0.01	1.01	1.00
31676860	Brain region volumes	b	rs4837580				T	C	0.39	−0.05	−0.03	−0.01	0.0005	0.00	1.00	1.01
31676860	Brain region volumes	b	rs10983204				T	C	0.39	−0.05	−0.03	−0.01	0.0005	0.00	1.00	1.01
31015462	Estimated glomerular filtration rate	b	rs13283416	G	0.50	6.93	G	T	0.426	0.11	0.07	0.00	−0.0008	0.00	0.99	0.99
30604766	Estimated glomerular filtration rate	b		G	0.50	0.23	G	T	0.426	0.11	0.07	0.00	−0.0008	0.00	0.99	0.99
30374069	Osteoarthritis (hip)	b		G		1.10	G	T	0.426	0.11	0.07	0.00	−0.0008	0.00	0.99	0.99
30224653	Diastolic blood pressure	b	rs1861881	G	0.68	0.12	T	G	0.316	0.18	0.13	−0.01	0.0001	−0.01	1.00	0.99
32632093	Migraine and/or diastolic blood pressure	b					T	G	0.316	0.18	0.13	−0.01	0.0001	−0.01	1.00	0.99
18839057	Attention deficit hyperactivity disorder	a	rs10983238				G	C	0.233	0.02	0.00	0.01	−0.0008	−0.01	0.99	0.99
30239722	Waist-to-hip ratio adjusted for BMI	b	rs17292540	C	0.22	0.01	C	G	0.233	0.02	0.00	0.01	−0.0008	−0.01	0.99	0.99
29781551	Axial length or spherical error (univariate decomposition analysis)		rs12340737				A	C	0.332	−0.12	−0.08	−0.01	0.0003	0.00	1.00	1.01
30595370	Waist-hip ratio	b	rs35910339				C	G	0.317	0.01	0.02	0.01	−0.0007	0.00	1.00	0.99
31669095	Waist-to-hip ratio adjusted for BMI	b	rs811458				T	C	0.317	0.01	0.02	0.01	−0.0007	0.00	1.00	0.99
30595370	Systolic blood pressure	b	rs811689				T	C	0.447	−0.16	−0.11	0.00	0.0005	0.01	1.01	1.02
33407418	Glucagon levels in response to oral glucose tolerance test (decremental area under the curve for 0–120 min)		rs719535	T	0.24	0.18	T	C	0.243	0.01	0.01	−0.01	0.0001	0.02	1.01	1.00
30038396	Educational attainment (years of education)	b	rs10983324	A	0.30	0.01	A	C	0.299	0.12	0.04	0.00	−0.0007	0.00	0.99	0.99
32279069	Attention deficit hyperactivity disorder (persistent)	b	rs4836899	T	0.52	1.12	C	T	0.461	0.10	0.05	−0.02	0.0001	−0.01	0.99	1.00
31530798	Caudal middle frontal gyrus volume	a	rs10116120				T	G	0.241	0.10	0.11	−0.03	−0.0002	−0.02	0.98	0.99
30718901	Depression	b,c	rs10817969	T	0.72	1.02	G	T	0.28	0.08	0.07	−0.03	−0.0002	−0.03	0.98	0.99
31926635	Bipolar disorder or major depressive disorder	b	rs10759881	A		1.03	C	A	0.277	0.06	0.06	−0.03	−0.0002	−0.02	0.98	0.99
29700475	Depression	b	rs7856424	C	0.71	1.03	T	C	0.283	0.08	0.08	−0.03	−0.0002	−0.02	0.98	0.99
25524916	Glucose homeostasis traits		rs7036846			10.86										
23251661	Obesity-related traits	a	rs16934284	G	0.09	0.03	C	T	0.108	0.00	0.04	−0.02	−0.0005	−0.02	0.98	0.98
20195266	Response to antipsychotic treatment	a	rs4838255		0.14		T	A	0.179	0.02	0.03	−0.02	0.0001	0.01	0.99	1.00
31043756	Bipolar I disorder	b	rs7858026	T	0.48	1.08	T	A	0.488	−0.04	−0.04	−0.02	−0.0001	0.00	1.00	1.00
29500382	Feeling guilty	b	rs35623509	C	0.27	5.61	G	C	0.275	0.08	−0.01	0.01	−0.0001	−0.02	1.00	1.00
33632238	Ischemic stroke in diabetes mellitus	a	rs189233549	G	0.01	3.91										
29500382	Neuroticism	b	rs579017	T	0.06	5.65	T	C	0.06	0.00	0.02	0.05	−0.0002	0.08	1.03	1.01
29942085	Neuroticism	b		T		5.47	T	C	0.06	0.00	0.02	0.05	−0.0002	0.08	1.03	1.01

Where: RA, risk allele; RAF, risk allele frequency; a, samples size < 1000; b, sample size > 10,000; c, replication analysis conducted; bold indicates significant at *p* < 5.45 × 10^−5^ in UK biobank.

**Table 4 genes-12-01194-t004:** Genotype-specific expression patterns in relevant tissues.

Signal	Query	Direction	EA	EAF	RSID	R2	Gene	Tissue	Non-Effect	Allele		NEA	EA	Size	*p*-Value
SBP/DBP-1	rs4837585	-	T	0.46	rs7863794	0.92	*PAPPA*	Muscle	-	Skeletal		A = 0.535	G = 0.465	0.164925	8.09 × 10^−6^
					rs811689	0.92	*PAPPA*	Muscle	-	Skeletal		C = 0.535	T = 0.465	0.161858	9.87 × 10^−6^
					rs7857286	0.92	*TRIM32*	Adipose	-	Visceral	(Omentum)	C = 0.535	T = 0.465	0.177953	8.28 × 10^−7^
					rs1661294	0.90	*TRIM32*	Adipose	-	Visceral	(Omentum)	G = 0.54	A = 0.46	0.175288	8.46 × 10^−7^
					rs10817910	0.92	*TRIM32*	Adipose	-	Visceral	(Omentum)	C = 0.535	A = 0.465	0.176971	1.04 × 10^−6^
					rs1885242	0.92	*TRIM32*	Adipose	-	Visceral	(Omentum)	G = 0.535	A = 0.465	0.176971	1.04 × 10^−6^
					rs9775101	0.92	*TRIM32*	Adipose	-	Visceral	(Omentum)	C = 0.535	T = 0.465	0.176971	1.04 × 10^−6^
DBP-2	rs34432054	T	+	0.13	rs34789583	1.00	*TLR4*	Whole	Blood			G = 0.874	A = 0.126	−0.19938	5.49 × 10^−16^
					rs35940453	1.00	*TLR4*	Whole	Blood			C = 0.874	T = 0.126	−0.19938	5.49 × 10^−16^
					rs7857333	1.00	*TLR4*	Whole	Blood			G = 0.874	A = 0.126	−0.19938	5.49 × 10^−16^
					rs13299033	1.00	*TLR4*	Whole	Blood			A = 0.874	T = 0.126	−0.19938	5.49 × 10^−16^
					rs35199804	1.00	*TLR4*	Whole	Blood			G = 0.874	T = 0.126	−0.19938	5.49 × 10^−16^
BMI-1	rs55654527	A	-	0.34	rs957512	0.98	*TLR4*	Whole	Blood			T = 0.621	C = 0.379	0.105626	4.81 × 10^−11^
					rs12001083	0.98	*TLR4*	Whole	Blood			C = 0.621	T = 0.379	0.105061	6.01 × 10^−11^
					rs10759926	1.00	*TLR4*	Whole	Blood			T = 0.616	C = 0.384	0.097862	1.05 × 10^−09^
					rs10116193	1.00	*TLR4*	Whole	Blood			A = 0.616	G = 0.384	0.097751	1.09 × 10^−09^
					rs10983720	1.00	*TLR4*	Whole	Blood			A = 0.616	T = 0.384	0.097751	1.09 × 10^−09^
WHRadjBMI-1	rs13283416	-	G	0.43	rs6478243	0.81	*ASTN2*	Adipose	-	Subcutaneous		T = 0.621	C = 0.379	−0.17614	1.72 × 10^−06^
Neuroticim-1	rs579017	+	T	0.06	rs579017	1.00	*TLR4*	Whole	Blood			T = 0.045	C = 0.955	0.205844	2.88 × 10^−11^
Mood-2	rs79825568	-	A	0.48	rs928052	1.00	*TLR4*	Artery	-	Tibial		G = 0.49	A = 0.51	−0.15591	4.98 × 10^−8^
					rs13294726	1.00	*TLR4*	Artery	-	Tibial		T = 0.49	C = 0.51	−0.15483	5.00 × 10^−8^
					rs12236328	1.00	*TLR4*	Artery	-	Tibial		A = 0.49	G = 0.51	−0.15608	5.68 × 10^−8^
					rs1887905	1.00	*TLR4*	Artery	-	Tibial		C = 0.49	G = 0.51	−0.15446	8.30 × 10^−8^
					rs13293271	1.00	*TLR4*	Artery	-	Tibial		A = 0.49	G = 0.51	−0.15418	8.40 × 10^−8^
					rs928052	1.00	*TLR4*	Whole	Blood			G = 0.49	A = 0.51	−0.16194	8.12 × 10^−25^
					rs13293271	1.00	*TLR4*	Whole	Blood			A = 0.49	G = 0.51	−0.16151	4.00 × 10^−24^
					rs1887905	1.00	*TLR4*	Whole	Blood			C = 0.49	G = 0.51	−0.16208	4.50 × 10^−24^
					rs913615	1.00	*TLR4*	Whole	Blood			C = 0.49	A = 0.51	−0.16069	8.92 × 10^−24^
					rs4481681	1.00	*TLR4*	Whole	Blood			A = 0.49	G = 0.51	−0.16069	8.92 × 10^−24^
					rs12236328	1.00	*TLR4*	Nerve	-	Tibial		A = 0.49	G = 0.51	−0.17896	9.92 × 10^−6^
					rs13293271	1.00	*TLR4*	Nerve	-	Tibial		A = 0.49	G = 0.51	−0.1801	8.64 × 10^−6^
					rs13294726	1.00	*TLR4*	Nerve	-	Tibial		T = 0.49	C = 0.51	−0.18186	6.19 × 10^−6^
					rs928052	1.00	*TLR4*	Nerve	-	Tibial		G = 0.49	A = 0.51	−0.18426	4.40 × 10^−6^
Anhedonia-1	rs144850429	+	T	0.01	rs144850429	1.00	*ASTN2*	Nerve	-	Tibial		C = 0.99	T = 0.01	−0.62966	1.81 × 10^−6^

## Data Availability

UK Biobank data is available upon application to the UK Biobank (https://www.ukbiobank.ac.uk/enable-your-research/register, accessed on 26 July 2021). Results of the analyses presented here are available upon request.

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
