# Peer review of "Genetic Variation in the ASTN2 Locus in Cardiovascular, Metabolic and Psychiatric Traits: Evidence for Pleiotropy Rather Than Shared Biology"

_genes, 2021, doi:10.3390/genes12081194_

Round 1

Reviewer 1 Report

This study concerns analysis of genetic variation in the ASTN2 locus in cardiometabolic and psychiatric traits. The methodology is based on data of 402111 British subjects from the UK Biobank, the methodological procedures (anthropometric, physiological measures, genetic analyses) are sufficiently described. The main funding is that ASTN2 has a pleiotropic effect on psychiatric and cardiovascular/metabolic traits. The authors critically discuss the findings in accordance with current studies. This study is interesting and well-written providing novel information in this research field.

I have only minor comments:

p.5, line 229 – The authors mentioned “poor metabolic health”. In this context, I recommend including more information about metabolic syndrome definition and clinical symptoms. Moreover, it is also important from the aspect of BMI and related eating disorders (as mental/psychiatric diagnosis).

I recommend including future research directions in a separate paragraph.

Author Response

Thank you for your suggestions, we have made the following changes:

  1. We have expanded on metabolic syndrome and its link with the brain and eating disorders in the results (lines 304-3010).
  2. We have added a paragraph on future work that should be considered, in the Discussion (lines 520-525).

Reviewer 2 Report

Thank you for providing this opportunity to review this paper.

In this manuscript, Burt et al. have investigated the association between genetic variation in the ASTN2 locus and the shared pathology of cardiometabolic and psychiatric illness. To address this, they have excavated the UK Biobank dataset (This is one of the most extensive studies to date with genetic data and a wide range of cardiometabolic, psychological, and psychiatric phenotyping. In this study, the authors have performed a systematic analysis in their research and shown that ASTN2 locus has pleiotropic effects on cardiometabolic and psychiatric traits rather than contributing to shared pathology. Furthermore, it was also demonstrated that the SNPs in this locus acts through ASTN2 to influence central fat accumulation and the risk of anhedonia. Additionally, The experiments were well organized, and the results are presented in systematically organized figures. The experimental results were well validated statistically. The manuscript presents essential aspects of investigation in the field. Hence, it was found to be suitable for publication. However, there are some minor concerns, and they are listed in the below comments. Please feel free to consider these comments for further decisions.

  1. The authors have reported the main limitations of their study: the cohort included in their study is biased towards healthier and wealthier individuals. Hence, their study presents only generalized results, for instance, the White British ancestry individuals. Also, a cohort with a higher prevalence of cardiometabolic and psychiatric conditions should be included in future studies, which would allow the identification of novel SNPs in the locus.
  2. There is no functional validation to support the in silico results. Towards this, the authors may consider an in-vo experiment to address this issue. In this regard, though there are no murine astn2 knockout models have been developed. A brinp1 null mice, which has possibly overlapping functions, as described by the authors (cited in this manuscript, 46), can be investigated. An in-vitro validation on the ASTN2 knockout neuron cells would be able to address the issue. This would help to elucidate the role of ASTN2 in this particular cell type.
  3. The author states that the loci for mood instability, neuroticism, blood pressure, and BMI might be acting through other genes in the region, supported by their effects on the mRNA expression of TRIM32 and TLR4. The experimental results to confirm this notion is lacking.

Author Response

Thank you for your suggestions, we have made the following changes:

  1. The missing parenthesis has been added on line 51.
  2. The requested information has been provided on line 69, along with the url to the UK Biobank protocol in case readers require further information.
  3. In Table 1 the footer explains that continuous variables are presented as mean and standard deviation, whereas binary data is presented as N (%). The first row has been amended for clarity and now specifies that here the parenthesis give the % of male participants in the group. These parentheses have been removed from the Men and Women columns as these do not add information, but it remains in the All group, as here it is useful information.

Where continuous variables are presented as mean (standard deviation) and binary variables as N (%). IMT, carotid intima-media thickness; ISH, ischemic heart disease; GAD, generalised anxiety disorder; MDD, major depressive disorder; BD, bipolar disorder. *only available for 22179 individuals.** only available for 128751 individuals.

  1. We have reformatted the figures so that all panels and the legend are on the same page for ease of reading.
  2. Figure 3C has been enlarged.
  3. The supplementary figures are now included in the main text, again with all panels and the legend on one page.

Reviewer 3 Report

The manuscript by Burt and colleagues, entitled "Genetic variation in theASTN2 locus in cardiovascular, meta- 2 bolic and psychiatric traits: evidence for pleiotropy rather than 3 shared biology" discusses the link between cardiometabolic and psychiatric diseases starting from the assumption that recent research highlight biological mechanisms shared with an important role of IlASTN2 locus.

I appreciate the authors' attempt to address this issue and the truly massive data analysis. The work is well written, clear and straightforward. And it represents a start to go deeper.

Given the premises, apart from some minor corrections, I encourage the publication of the manuscript.

  • Line 51: a parenthesis is opened which then does not close “(SNPs ….”

  • Line 69: “At baseline, a blood sample was 69 collected and stored appropriately for DNA extraction” The blood was stored how? Whole, plasma or serum? And at what temperature?

  • Page 3, Table 1: The authors write in the legend that the variables are presented as mean (standard deviation). However, in the first row of the table (N (% bad) 184861 (100.0) 217250 (0.0) 402111 (46.0)) we talk about the number of individuals, so there are no SDs. So, the digits between brackets, what do they represent? Also, what does % males mean?

  • Figure 1: Authors are requested to put the legend immediately above/below the figure. In addition, authors are invited to restore the page numbers that have evidently been lost after the rotation of the pages. This figure consists of 7 panels. All panels must be included on a single page.

  • Figure 3: The authors are asked to enlarge panel C. Unfortunately, the legend is very difficult to read and by enlarging the figure, it grains a lot.

  • Supplementary figures: Also in this case, if a figure is made up of tot panels, these should be paginated on a single page. Furthermore, I invite the authors to consider incorporating the supplementary figures into the main text.

Author Response

Thank you for your suggestions, we have made the following changes:

  1. The need to expand the study to higher psychiatric and cardiometabolic risk individuals has been noted in the Discussion (lines 520-525)
  2. Unfortunately in vivo analysis is beyond the scope of this report. However, this point has been highlighted in the paragraph on further work (line 520-525).
  3. The possibility of genetic variants acting through TRIM23 or TLR4 is based upon the lead SNPs for these traits demonstrating genotype-specific expression of these genes in relevant tissues, but not ASTN2. We have clarified this by referring the reader to Table 4 in the discussion as well as in the results (line 484).